# Evaluating spatially adaptive guidelines for the treatment of gonorrhea to reduce the incidence of gonococcal infection and increase the effective lifespan of antibiotics

**Reza Yaesoubi**[1]*, **Ted Cohen**[2], **Katherine Hsu**[3], **Thomas L. Gift**[4], **Sancta B. St. Cyr**[4], **Joshua A. Salomon**[5], **Yonatan H. Grad**[6]

**1** Department of Health Policy and Management, Yale School of Public Health, New Haven, Connecticut, United States of America, **2** Department of Epidemiology of Microbial Diseases, Yale School of Public Health, New Haven, Connecticut, United States of America, **3** Massachusetts Department of Public Health, Boston, Massachusetts, United States of America, **4** Division of STD Prevention, Centers for Disease Control and Prevention, Atlanta, Georgia, United States of America, **5** Department of Health Policy, Stanford University School of Medicine, Palo Alto, California, United States of America, **6** Department of Immunology and Infectious Diseases, Harvard T. H. Chan School of Public Health, Boston, Massachusetts, United States of America

* reza.yaesoubi@yale.edu

**Data Availability Statement:** The data underlying the results presented in the study are available from the Gonococcal Isolate Surveillance Project

## Abstract

In the absence of point-of-care gonorrhea diagnostics that report antibiotic susceptibility, gonorrhea treatment is empiric and determined by standardized guidelines. These guidelines are informed by estimates of resistance prevalence from national surveillance systems. We examined whether guidelines informed by local, rather than national, surveillance data could reduce the incidence of gonorrhea and increase the effective lifespan of antibiotics used in treatment guidelines. We used a transmission dynamic model of gonorrhea among men who have sex with men (MSM) in 16 U.S. metropolitan areas to determine whether spatially adaptive treatment guidelines based on local estimates of resistance prevalence can extend the effective lifespan of hypothetical antibiotics. The rate of gonorrhea cases in these metropolitan areas was 5,548 cases per 100,000 MSM in 2017. Under the current strategy of updating the treatment guideline when the prevalence of resistance exceeds 5%, we showed that spatially adaptive guidelines could reduce the annual rate of gonorrhea cases by 200 cases (95% uncertainty interval: 169, 232) per 100,000 MSM population while extending the use of a first-line antibiotic by 0.75 (0.55, 0.95) years. One potential strategy to reduce the incidence of gonorrhea while extending the effective lifespan of antibiotics is to inform treatment guidelines based on local, rather than national, resistance prevalence.

## Author summary

Antimicrobial resistance threatens the effective treatment of a growing number of infections worldwide. In the absence of rapid point-of-care diagnostics that determine

(https://www.cdc.gov/std/gisp/default.htm) and Sexually Transmitted Disease Surveillance 2019 (https://www.cdc.gov/std/statistics/2019/default. htm). The code of our simulation model and analyses are publicly available without restriction through GitHub: https://github.com/yaesoubilab/APACEc.

**Funding:** This work was supported by the U.S. Centers for Disease Control and Prevention (CDC), National Center for HIV, Viral Hepatitis, STD, and TB Prevention Epidemiologic and Economic Modeling Agreement (5NU38PS004644, https:// www.cdc.gov/nchhstp/neema) to JAS and by R01AI153351 from the National Institute of Allergy and Infectious Diseases (https://www.niaid.nih. gov/) to RY. The Centers for Disease Control and Prevention contributed to study design and preparation of the manuscript. YHG was supported by R01AI132606 and R01AI153521, and TC by R01AI112438, all from the National Institute of Allergy and Infectious Diseases (https://www.niaid. nih.gov/). The National Institute of Allergy and Infectious Diseases had no role in study design, data collection and analysis, decision to publish, or preparation of the manuscript.

**Competing interests:** I have read the journal's policy and the authors of this manuscript have the following competing interests: RY, TC, JAS, and YHG received funding from the National Institute of Health and the U.S. Centers for Disease Control and Prevention. TLG and SBS are employed by the U.S. Centers for Disease Control and Prevention. YHG has received research funding from Pfizer and Merck for unrelated work and has received consulting fees from GSK.

antibiotic susceptibility, the treatment of several infections caused by bacteria (e.g., gonorrhea and tuberculosis) remain empiric and informed by guidelines. These guidelines are usually determined at the national level and based on the estimated resistance prevalence nationally. Here, we show that more cases of gonorrhea could be prevented and the effective lifespan of antibiotics suitable for the treatment of gonorrhea could be extended if treatment guidelines are determined locally and based on the regional resistance prevalence rather than on a single nationwide recommendation. Our analysis provides evidence to highlight the importance of 1) maintaining local surveillance systems of antimicrobial resistance and 2) engaged policymakers who use the data from these surveillance systems to inform timely and effective decisions at the local level.

## Introduction

Gonorrhea is a major public health concern, with 583,405 reported cases in 2018 in the United States [1] and an estimated 87 million cases worldwide in 2016 [2]. In both 2013 and 2019, the U.S. Center for Disease Control and Prevention (CDC) named antimicrobial-resistant (AMR) gonorrhea among the most urgent infection threats in the United States, as *N. gonorrhoeae*, the bacterial pathogen that causes gonorrhea, has developed resistance to all antibiotics used to treat it [3]. The threat of untreatable gonococcal infections highlights the need for strategies to reduce the burden of gonorrhea and maximize the clinically useful lifespan of existing antibiotics while awaiting the introduction of new anti-gonococcal antibiotics [4,5].

The diagnosis of gonorrhea is usually made by nucleic acid amplification test, and treatment is most often empiric and based on national guidelines [3,6–8]. Even when culture is available, patients usually receive first-line empiric antibiotic treatment while awaiting drug-susceptibility results. In the US, guidelines for the treatment of gonorrhea are determined nationally and based on the prevalence of antimicrobial resistance estimated by the Gonococcal Isolate Surveillance Project (GISP) [9]. GISP is a sentinel surveillance system that monitors trends in antimicrobial susceptibilities of gonococcal strains in the US [10]. To ensure the effectiveness of first-line empiric treatment, only antibiotics with low prevalence of resistance are considered for use as first-line therapy. Historically, the World Health Organization (WHO) recommended switching the first-line antibiotic gonorrhea treatment once the prevalence of resistance to that antibiotic exceeds 5% [9,11].

The data from GISP reveal heterogeneity in the prevalence and trends of AMR gonorrhea across the participating surveillance sites. For example, in 2017, although overall 30% of *Neisseria gonorrhoeae* isolates collected through GISP were resistant to ciprofloxacin, this percentage varied widely (2.5% to 56%) across GISP surveillance sites [12]. This suggests that guidelines for the first-line therapy of gonorrhea that are based on a national average estimate of resistance prevalence may not provide optimal treatment recommendations for all regions. For regions with higher than average prevalence of resistance to first-line antibiotics, a greater proportion of gonorrhea cases will not receive an antibiotic that matches their susceptibility profile. This would lead to longer durations of infectiousness, facilitating further transmission of resistant gonorrhea. In contrast, for regions with lower than average prevalence of resistance to first-line antibiotics, revising the national guidelines prematurely will remove an effective antibiotic from clinical use and would lead to earlier and more extensive use of second-line regimens. This suggests that one potential strategy to extend the effective lifespan of existing antibiotics is to optimize local treatment guidelines using local resistance prevalence rather than relying on a single national recommendation.

In this study, we used a transmission dynamic model to compare the performance of policies that inform guidelines for first line gonococcal therapies. Specifically, we evaluated the degree to which locally tailored recommendations could extend the effective lifespan of antibiotics compared with the current switching strategy, which is based on the national estimate of resistance prevalence from annually reported surveillance efforts. We also investigated how increasing the frequency and size of drug-resistance surveys would impact the performance of these strategies to inform guidelines for the first-line therapy of gonococcal infections.

## Methods

### Simulating guidelines for the first-line treatment of gonorrhea

We developed a transmission dynamic model of gonorrhea to compare the performance of four types of strategies to generate first-line treatment recommendations in terms of their ability (a) to prolong the effective life of antibiotics and (b) to reduce the incidence of gonorrhea (Table 1). The 'Base' strategy represents the guideline that recommends switching to a new first-line antibiotic once the national estimate of resistance prevalence during the past year passes a certain threshold. A threshold of 5% in the 'Base' strategy is consistent with the historical recommendations of the WHO, as the estimates of resistance prevalence from surveillance systems (such as GISP in the United States) become available on yearly basis [9,11]. Consistent with the observation that 5,061 GISP isolates were tested for drug susceptibility in 2017 with an average of 196 annual tests for each surveillance site [12], we assumed that for the 'Base' strategy, 200 isolates are tested annually in each area.

The 'Spatial' strategy seeks to delay the emergence of resistance to the first-line treatment regimen by using regional estimates of resistance prevalence to inform each region's recommended regimen. The strategy 'Enhanced Spatial' is similar to 'Spatial' except that to obtain local estimates of resistance prevalence, it assumes 800 susceptibility tests each year (as opposed to 200). With 200 susceptibility tests, the local prevalence of resistance can be estimated with a 95% Wilson's confidence interval of 2.7%-9.0% if the true prevalence of resistance reaches 5%. This confidence interval can be reduced to 3.7%-6.7% when 800 susceptibility tests are performed.

The fourth strategy, 'Enhanced Spatial-Quarterly', differs from 'Enhanced Spatial' in how frequently the estimates of resistance prevalence are obtained and treatment recommendations are updated. The 'Enhanced Spatial-Quarterly' strategy relies on the same annual number of susceptibility tests as in 'Enhanced Spatial,' but assumes that survey data are reported at 3-month intervals. Therefore, the quarterly strategy may detect changes in the point prevalence of resistance more quickly, but at the expense of lowering the precision in these estimates.

**Table 1. Adaptive guidelines to inform first-line treatment recommendations for gonorrhea.**

| Strategies | Frequency of Decision Making | Annual Number of Tests for Resistance at Each Surveillance Site[‡] | Policy Examples |
|---|---|---|---|
| Base | Annually | 200 | Switch to a new first-line drug in all metropolitan areas when the estimate for the national proportion of resistant isolates during the past year exceeds $x$%. |
| Spatial | Annually | 200 | Switch to a new first-line drug only in metropolitan areas with the estimate for the proportion of resistant isolates during the past year exceeds $x$%. |
| Enhanced Spatial | Annually | 800 | Same as 'Spatial' |
| Enhanced Spatial-Quarterly | Quarterly | 800 | Same as 'Spatial' |

[‡] In 2017, 5,061 GISP isolates were tested for drug susceptibility with an average of 196 annual tests for each surveillance site [12].

We simulate a scenario in which three hypothetical antibiotics (Drug A, Drug B, and Drug M) are available for the treatment of gonococcal infection. Drug A represents a first-line antibiotic, such as ceftriaxone [13,14], and Drug B represent an alternative antibiotic suitable for first-line treatment of gonorrhea, such as zoliflodacin [4] or gepotidacin [5], both of which have been over 95% effective against urogenital gonococcal infections in phase 2 trials. Drug M represents the last-line antibiotic that is reserved for use for those resistant to other current empiric regimens. Therefore, its use for the first-line treatment of gonorrhea should be delayed as long as possible.

We assumed Drug A is initially used for the empiric treatment of gonorrhea and that Drug B is used as the second-line therapy for cases that fail the treatment with Drug A. As more cases of gonorrhea are treated with Drug A, the selective pressure for resistance to Drug A increases. Under the current recommendations, Drug A would then be replaced with Drug B once the resistance to Drug A exceeds the 5% threshold [9,11]. Subsequently, those who fail first-line treatment with Drug B will be retreated with Drug M. Similarly, when the estimate for the prevalence of resistance to Drug B reaches a certain threshold, Drug B will be removed from the first-line therapy and Drug M will be used for both first-line and second-line therapy.

Our model describes the transmission of *N. gonorrhoeae* among men who have sex with men (MSM) in 16 metropolitan areas in the United States (Fig 1). We selected these 16 areas based on the availability of local data on the population of MSM and on the incidence of gonorrhea among MSM (Table A in S1 Text). Data from GISP clinics suggest approximately 37.2% of gonorrhea cases in 2018 were among MSM and the emergence of resistance among this population is of serious concern [1].

Our model is adapted from a prior study [15] with additional extensions to simulate the transmission of *N. gonorrhoeae* among the MSM population of 16 metropolitan areas. We assume no intercity transmission which implies that an individual's risk of infection depends

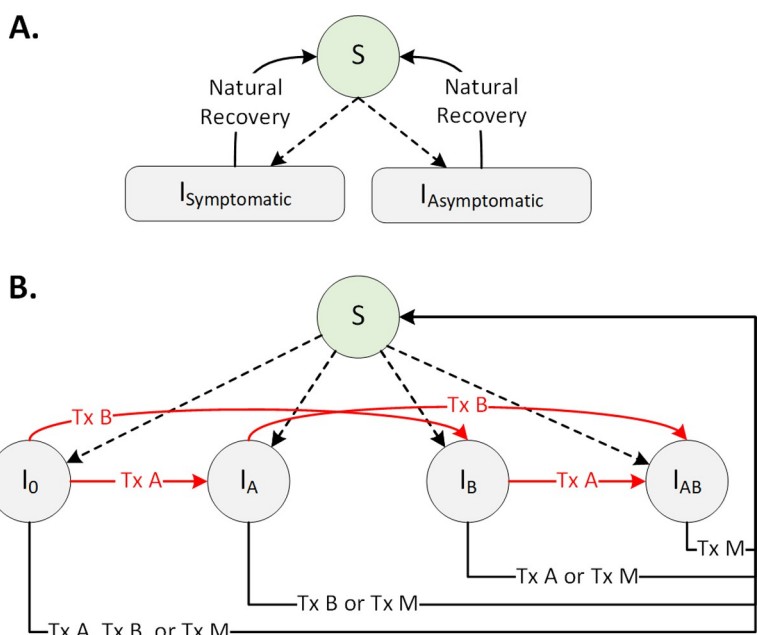

**Fig 1. A stochastic compartmental gonorrhea transmission model (adapted from [15] and extended to simulate the transmission of *N. gonorrhoeae* among men who have sex with men in 16 metropolitan areas in the United States).** Dotted arrows represent new infection and red arrows represent resistance acquisition while under treatment. S represents susceptible individuals, $I_0$ represents individuals with drug-susceptible infections, and $I_A$, $I_B$, and $I_{AB}$ represent individuals with infections resistant to Drug A, B, and both. Tx A, Tx B, and Tx M denote treatment with drugs A, B, and M.

on the local prevalence of infection in the metropolitan area where they reside (and is not impacted by the prevalence of infection in other metropolitan areas).

Infected cases can be symptomatic or asymptomatic (Fig 1A). Infected individuals are further classified by the resistant profile of the infecting strain: drug-susceptible infection ($I_0$), infection resistant to Drug A ($I_A$), infection resistant to Drug B ($I_B$), and infection resistant to both Drugs A and B ($I_{AB}$) (Fig 1B). Asymptomatic cases do not seek treatment and remain infectious until they recover spontaneously or get detected through active screening (Fig 1A). All symptomatic cases are seeking treatment with some delay.

Infected individuals who seek treatment or are detected through screening will receive treatment with either Drug A, B, or M, depending on the current recommendation for first-line therapy. We assume that treatment always fails if the infecting strain is resistant to the prescribed antibiotic. Treatment with an antibiotic to which the infecting strain is susceptible may lead to one of two possible outcomes: 1) the treatment succeeds and the individual returns to the susceptible state, or 2) the treatment leads to resistance and fails (Fig A in S1 Text). A fraction of symptomatic individuals failing the first-line treatment (due to receiving ineffective treatment or developing resistance) will seek retreatment with some delay. These individuals will receive a second-line antibiotic, which is Drug B in the absence of resistance to Drug B, and Drug M in the case of resistance to Drug B. We assume that treatment with Drug B could lead to the selection of resistance to Drug B with a small probability, upon which the individual will be treat with Drug M (Fig A in S1 Text). As soon as effective treatment (i.e., treatment with an antibiotic that matches the individual's susceptibility profile) is initiated, we assume that infected individuals no longer contribute to the force of infection (due to either negligible infectiousness or reduced sexual activity). Those symptomatic individuals who fails the first-line treatment but do not seek retreatment will remain in their current state where they may naturally recover or seek treatment in future time steps. Asymptomatic individuals may be detected only through screening.

Antibiotic treatment may select for resistance (represented by red arrows in Fig 1B), and resistant strains may be transmitted directly to other susceptible individuals. To account for fitness costs associated with resistance, we assumed that resistant strains are less transmissible than susceptible strains, at least initially [16,17]. Data from GISP indicate that despite tetracycline, penicillin, and ciprofloxacin no longer being used for gonococcal treatment, the prevalence of resistance to tetracycline and penicillin has been fairly stable and resistance to ciprofloxacin has increased [18]. To produce simulated trajectories that allow for this persistence despite reduced use of these antibiotics, we allow the fitness cost of resistance to gradually decrease, consistent with the idea that the fitness costs may be ameliorated through compensatory mutation [16]. We also allowed for the importation of resistant cases to each metropolitan area that occurs continuously over time according to a Poisson process. Additional details about the model are provided in S1 Text.

## Model calibration and validation

We used a Bayesian approach to calibrate our model jointly against the rate of reported gonorrhea for each metropolitan area in 2017 and against the overall estimates for the prevalence of gonorrhea and the proportion of gonorrhea cases that are symptomatic. We chose this calibration approach because estimates for prevalence of gonorrhea and the proportion of gonorrhea cases that are symptomatic were not available for each metropolitan area. This Bayesian approach seeks to estimate the probability distributions of unknown parameters such that trajectories that are simulated using random draws from these distributions fit the available epidemiological data [19].

As Drug A and Drug B are hypothetical future antibiotic treatments for gonorrhea, we did not specify prior distributions for the parameters that relate to the emergence and spread of resistance to these drugs. Instead, we derived relevant ranges for these parameters by retaining only those simulated trajectories where the prevalence of resistance to Drug A and Drug B reaches at least 5% in during the simulation (Tables C, E and F in S1 Text). This is motivated by the observation that this level of resistance has been observed for each of the past first-line antibiotics used to treat gonorrhea [1]. The details of the calibration procedure are provided in §S3 of S1 Text.

## Comparing the performance of treatment guideline strategies

We compared the performance of strategies in Table 1 based on the number of gonorrhea cases that are averted with respect to the status quo (the "Base" strategy in Table 1 with 5% switch threshold) and the increase in the effective life of Drugs A and B. We define the effective lifespan of Drugs A and B as the area under the curve of the percentage of gonorrhea cases that are successfully treated with Drugs A or B over 50 years of simulation. This performance measure is calculated as $\sum_{t=1}^{50} \frac{Z_A(t)+Z_B(t)}{Z_A(t)+Z_B(t)+Z_M(t)}$, where $Z_A(t)$, $Z_B(t)$ and $Z_M(t)$ are the number of gonorrhea cases treated successfully with Drugs A, B, or M during the simulation year $t$ [15] (Fig E in S1 Text).

As we will demonstrate later, changing the switch threshold from 5% would impact both the effective lifespan of Drugs A and B and the number of annual gonorrhea cases. Therefore, we define $\Delta E$ as the maximum increase in the effective lifespan of antibiotics that could be achieved compared to the status quo while keeping the gonorrhea incidence unchanged. For any strategy that extended the effective lifespan of Drugs A and B by $\Delta E$ years, we estimated the expected number of additional cases of gonorrhea that would be treated successfully with first-line antibiotics without increasing the number of gonorrhea cases during the simulation period with $V_0 \frac{\Delta E}{E_0}$, where $V_0$ is the total number of cases successfully treated with Drugs A or B during the simulation period, and $E_0$ is the effective lifespan of Drugs A and B under the status quo [15].

The simulation window of 50 years was selected to ensure enough time for the resistance against Drug A and Drug B to reach 5% (in a sensitivity analysis, we set the simulation window at 35 years). We summarized results using the mean and 95% uncertainty interval (i.e. the interval between 2.5th and 97.5th percentiles of realizations) across 500 simulated trajectories.

## Sensitivity analyses

Through a number of sensitivity analyses, we investigated the robustness of our conclusions with respect to the choice of parameters relating to the emergence and spread of resistance to Drugs A and B, the simulation duration, choice of prior distributions for model parameters, the initial prevalence of resistance to drug B, and the trend in the rate of reported gonorrhea cases and the prevalence of gonorrhea. Details of these sensitivity analyses are described in §S4 of S1 Text.

## Results

We fit our model to gonorrhea prevalence, the rate of reported gonorrhea cases in 2017 for all metropolitan areas included in the model, and the proportion of gonorrhea cases with symptoms (Fig 2A–2C). We estimated the proportion of cases resistant to Drugs A, B, or both A and B when 200 annual gonorrhea cases are tested for drug resistance at each surveillance site during each simulation (Fig 2D–2F). We note that trajectories displayed in Fig 2D–2F differ

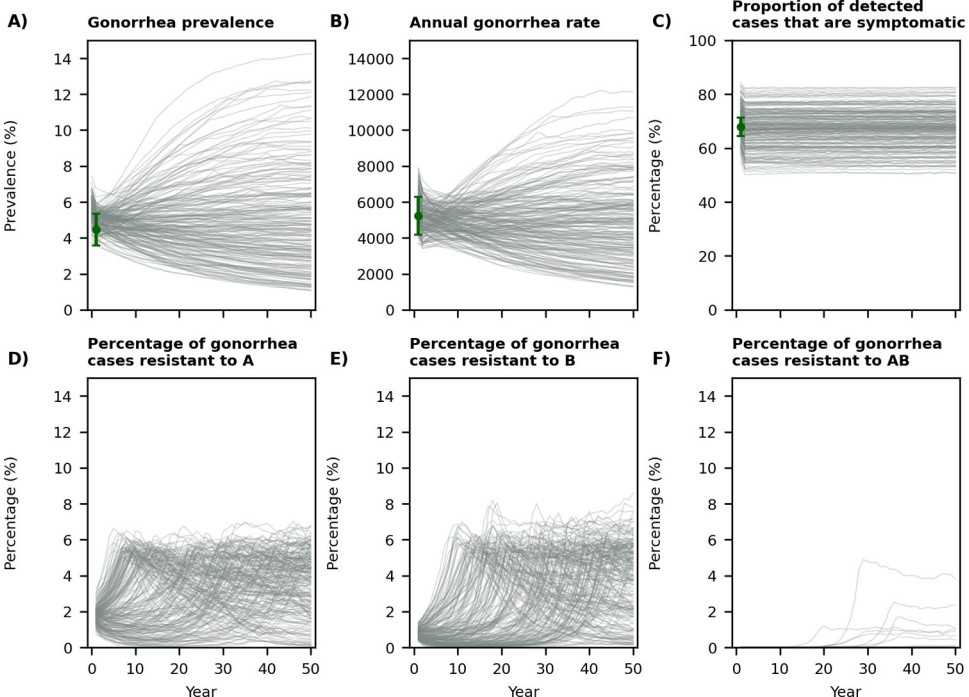

**Fig 2. Displaying 200 simulated trajectories from the calibrated model.** The green dots in panels A-C represent the estimates the model is calibrated against: gonorrhea prevalence (4.5% [3.6%, 5.4%] among MSM [28]), the estimated rate of reported gonorrhea cases in metropolitan areas included in our model in 2017 (5,548 cases per 100,000 MSM, Table A in S1 Text), and the proportion of gonorrhea cases among MSM that are symptomatic (67.9% [64.4–71.4%] [29]). In these simulated trajectories, the first-line treatment is changed when more than 5% of the annual gonorrhea cases are resistant to the first-line drug.

substantially in terms of the speed at which resistance emerges and spread. This is consistent with the history of *N. gonorrhoeae* resistance; for example, resistance to sulfonamides took only a few years to emerge and spread while resistance to penicillin took about 30 years to emerge and spread [20].

The fit of the model against the estimated rate of reported gonorrhea cases and the prevalence of infection among the MSM population of the metropolitan areas included in our model and the estimated proportion of cases resistant to Drugs A, B, or both in each area are shown in Fig C-Fig F of S1 Text. Fig 3 demonstrates that the rate at which resistance to Drug A increases within the 16 metropolitan areas of our model is consistent with the geographical variation in the trends of ciprofloxacin-resistant gonorrhea observed in GISP. We note that ciprofloxacin is no longer recommended as first-line therapy of gonorrhea. However, since the percentage of GISP isolates that exhibit resistance to ceftriaxone, the currently recommended first-line therapy for gonorrhea [14], is low (around 0.2% between 2014–2018 [12]), we used the historical estimates of the prevalence of ciprofloxacin resistance (Fig 3) to validate our model. The spatial heterogeneity in resistance in our model is driven by the difference in the rates of gonorrhea cases (Fig C in S1 Text), the prevalence of gonorrhea (Fig D in S1 Text), and the initial prevalence of resistance to Drugs A and B (Table E in S1 Text) across different metropolitan areas. This leads to different transmission dynamics in different metropolitan areas and hence, the resistance spreads at a different speed in different areas.

Our calibration approach reduced the uncertainty in the estimates for the following parameters (and the correlation between them): transmission parameter, probability that an infection

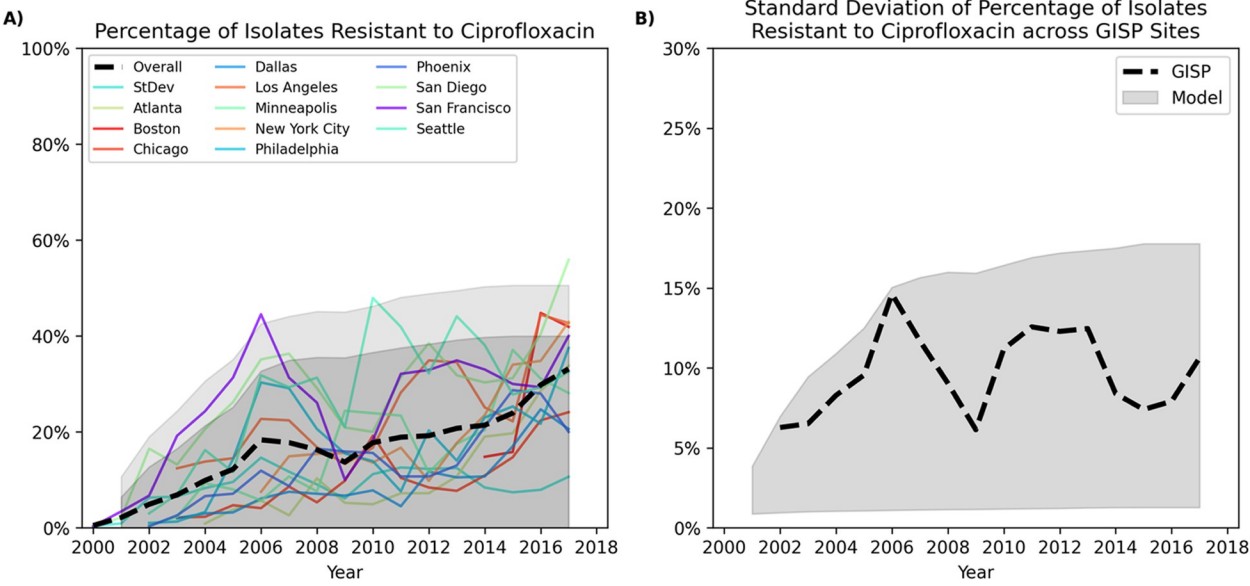

**Fig 3. Percentage of *Neisseria gonorrhoeae* isolates that are ciprofloxacin-resistant, overall and by GISP sites between 2000–2017 [12].** The shaded areas in Panel A represent the 95th (dark grey) and 99th (light grey) percentile interval of the prevalence of resistance to Drug A in the 16 metropolitan areas of our model. Panel B displays the standard deviation for the percentage of isolates resistant to ciprofloxacin across GISP sites. The shaded area represents the 95th percentile interval for the standard division of the prevalence of resistance to Drug A in the 16 metropolitan areas of our model. This figure demonstrates that the rate at which resistance to Drug A increases within the 16 metropolitan areas of our model is consistent with the geographical variation in the trends of ciprofloxacin-resistant gonorrhea observed in GISP.

will be symptomatic, duration of infections (without treatment), and time until screened for infection (Table B and Fig G in S1 Text). Based on the correlation between parameter values and the estimated lifespan of Drugs A and B, the following parameters had important effects: transmission parameter, duration of infections (without treatment), time until screened for infection, probability that an infection will be symptomatic, and parameters related to the fitness cost of Drugs A and B (Tables I-J in S1 Text).

Increasing the resistance-prevalence threshold for the 'Base' strategy (moving toward top-right corner of Fig 4A) increased the effective lifespan of Drugs A and B, i.e., allowed the use of these drugs for a longer period. Increasing this switching threshold, however, led to increases in the annual gonorrhea cases, since delaying the switch to a new antibiotic drug lowers the probability of receiving effective first-line therapy and extends the duration of infectiousness.

At the switch threshold of 5% and compared to the status quo, the 'Spatial' strategy could prevent an additional 200 (95% uncertainty interval: 169, 232) cases of gonorrhea per year per 100,000 MSM population and extend the effective lifespan of Drugs A and B by 0.75 (0.55, 0.95) years. If the switch threshold is further increased, the 'Spatial' strategy could increase the effective lifespan of Drugs A and B by 2.67 years without increasing the number of gonorrhea cases (this is measured as the horizontal distance between the points where the curves in Fig 4A crosses the x-axis). This is equivalent to successfully treating an additional 269 (157, 395) gonorrhea cases per 100,000 MSM population each year with Drugs A and B without worsening the burden of gonorrhea.

The benefits of the 'Spatial' strategy were enhanced under conditions that assumed a larger number of gonorrhea isolates could be tested for drug susceptibility to estimate the local prevalence of resistance with greater precision (Fig 4B). Compared to the status quo, a strategy that allows for local responses and performs 800 annual drug susceptibility tests in each area increased the effective lifespan of Drugs A and B by 2.89 years without increasing the annual

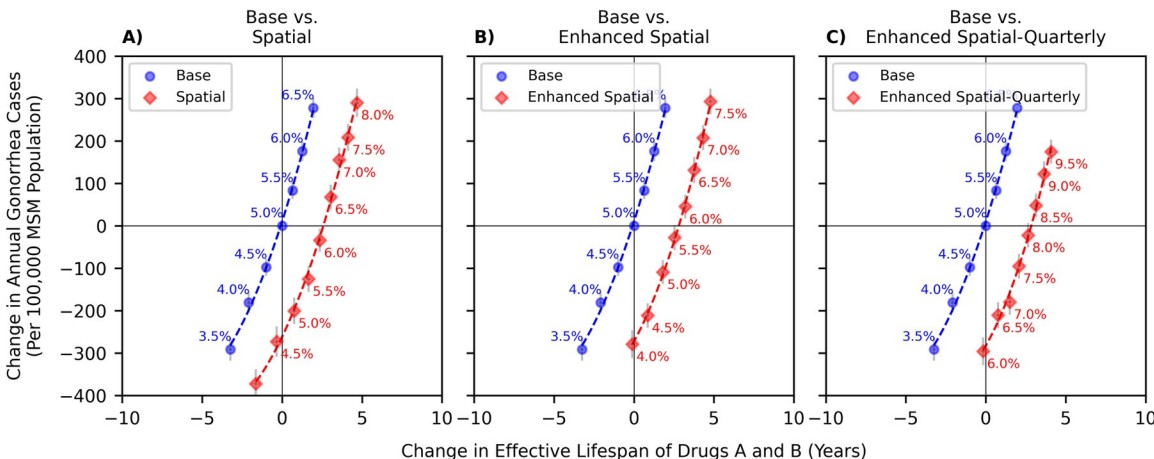

**Fig 4. Comparing the performance of policies in Table 1 with respect to the current policy.** The origins in these figures reflect the current policy that recommends switching the antibiotic used for empiric treatment once the estimated resistance prevalence exceeds 5% [9,11]. The numbers on the curves represent the threshold of resistance prevalence to switch the first-line therapy of gonorrhea. The bars represent 95% confidence intervals. As we are assuming that the decisionmaker is attempting to maximize the *expected* lifespan of antibiotics and minimize the *expected* number of gonorrhea cases, we are reporting estimated means and confidence intervals in this figure to show that the difference in the performance of these strategies are statistically significant. We note that the horizontal confidence intervals are difficult to see because they are contained in the markers.

incidence of gonorrhea. This is equivalent to successfully treating an additional 291 (170, 428) gonorrhea cases per 100,000 MSM population each year with Drugs A and B without worsening the burden of gonorrhea. At the switch threshold of 5%, this strategy could prevent an additional 28 (-2, 58) cases of gonorrhea per year per 100,000 MSM population and extend the effective lifespans of Drugs A and B by 2.54 (2.38, 2.69) years.

Increasing the frequency at which first-line therapy recommendations are revisited also led to an additional increase in the effective lifespan of Drugs A and B without increasing the number of gonorrhea cases (Fig 4C). Compared to the status quo, a strategy that determines the treatment guidelines locally, revisits them quarterly, and performs 800 annual drug susceptibility tests in each area (as opposed to 200) increased the effective lifespan of Drugs A and B by 2.93 years (equivalent of successfully treating an additional 295 (172, 434) gonorrhea cases per 100,000 MSM population each year with Drugs A and B) without increasing the burden of gonorrhea. For this strategy to improve both the number of gonorrhea cases averted and the effective lifespan of antibiotics, the switch threshold should be increased from 5% (Fig 4C). When guidelines are revisited quarterly (instead of annually), the 5% switch threshold triggers the switch to the new antibiotic too early. Therefore, while it could improve the number of gonorrhea cases averted, it also diminishes the effective lifespan of Drugs A and B. By increasing the switch threshold (to say, 7%), both outcomes could be improved under this strategy (Fig 4C).

While the rate of reported gonorrhea cases among the MSM population has increased significantly over the past few years [1], our model was calibrated to a constant incidence and prevalence (Fig 2A and 2B). To evaluate the sensitivity of our conclusions to the increase in rate of gonorrhea cases, we also modelled a scenario where the value of the transmission parameter is increased by 10%, which led to simulation trajectories with an increasing rate of gonorrhea and prevalence of infection (Fig L in S1 Text). Our conclusions are not meaningfully altered under this scenario (Fig M in S1 Text). The results of other sensitivity analyses are presented in §S4 of S1 Text.

## Discussion

In the absence of a point-of-care diagnostic test that determines antibiotic susceptibility, empiric treatment of gonorrhea remains standard practice. Empiric treatment guidelines are determined at the national level based on estimates of resistance prevalence provided by population surveillance. According to WHO recommendations, an antibiotic with resistance prevalence exceeding 5% should be considered for replacement in the guidelines by a new antibiotic with low level of population-wide resistance [9,11]. Using a transmission dynamic model of gonorrhea among the MSM population of 16 U.S. metropolitan areas, we projected how alternative strategies to inform the first-line treatment recommendation would impact the effective lifespan of antibiotics and the incidence of gonorrhea compared with standard practice.

Our analysis suggests that the choice for the threshold prevalence of resistance at which to switch to a new antibiotic requires a tradeoff between the effective lifespan of antibiotics and the incidence of gonorrhea. A higher threshold extends the effective lifespan of in-use antibiotics by delaying their removal but reduces the probability that an infected individual receives effective empiric therapy. This could facilitate further transmission of gonorrhea and lead to increased disease incidence. In contrast, a lower threshold prevents more gonorrhea cases by increasing the probability of effective empiric therapy at the expense of the antibiotic's effective lifespan.

In a recent study [15], we showed that the effective life span of antibiotics could be extended without worsening the burden of gonorrhea if treatment guidelines are revised 1) more frequently (quarterly vs. annually) or 2) based on both the estimated prevalence of resistance and the trend in the prevalence of resistance. The analysis here identifies an alternative strategy to extend the effective lifespan of antibiotics. We showed that compared to the current strategy that uses the national estimate of resistance prevalence to determine treatment guidelines, revising guidelines based on local estimates of resistance prevalence could extend the effective lifespan of antibiotics while reducing the burden of gonorrhea. We also showed that the reduction in the cases of gonorrhea and the increase effective lifespan of antibiotics can be augmented if the estimates of resistance prevalence are based on a larger number of drug-susceptibility tests and are used more frequently (quarterly vs. annually) to update treatment guidelines.

The improved performance of spatially-adaptive strategies to inform treatment guidelines can be attributed to the substantial differences in the prevalence of AMR gonorrhea across different regions (Fig 3). Accordingly, a strategy that determines national treatment guidelines based on the overall estimate of resistance prevalence may not provide optimal treatment recommendations for all local regions. A strategy based on the national average level of resistance is expected to result in delayed switching and possible increases in gonorrhea incidence in regions with higher prevalence of resistance to the empiric treatment and is expected to prematurely replace an effective empiric regimen in regions with lower low prevalence of resistance.

Our analysis has a number of limitations. First, our simulation model describes the spread of *N. gonorrhoeae* only among men who have sex with me (MSM) in 16 metropolitan areas of the United States. Compared to heterosexual men and women, the prevalence of gonorrhea and AMR gonorrhea is particularly high among MSM [6,18]. Therefore, the benefits of spatially-adaptive strategies might be lower for populations with lower burdens of gonococcal disease and AMR since the consequences of making suboptimal decisions would be less severe. Additional analysis of the advantages of spatially-adaptive strategies in this population, particularly as rates of gonorrhea and resistance rise, will be important to inform policy. Second, data on gonorrhea prevalence were only available for five metropolitan areas (Houston,

Miami, New York, San Francisco, and Washington, DC) (Fig D in S1 Text). For each simulated trajectory, we estimated the initial prevalence of gonorrhea in areas without locally available prevalence data by random draws from appropriate probability distributions (Table D in S1 Text). While this allowed us to confirm that our conclusions were robust over a range of plausible prevalence levels, limited local prevalence data precludes precise prediction of the magnitude of benefits of spatially-adaptive policies.

Third, we found estimates for the reported number of gonorrhea cases among the MSM population of only 3 of the 16 cities included in our model (New York City, Philadelphia, and San Francisco) [21]. For the remaining 13 cities, we assumed that the proportion of all gonorrhea cases attributable to MSM is the same for each location (Table A in S1 Text). Despite this assumption, we note that the calculated number of reported gonorrhea cases among the MSM population of cities considered here (i.e., 5,285 per 100,000 MSM, Table A in S1 Text) is consistent with the estimate of 5,241.8 per 100,000 MSM provided by the CDC's Sexually Transmitted Disease Surveillance 2018 report [1]. Fourth, we did not model specific antibiotics and instead chose to model hypothetical drugs with properties similar to antibiotics commonly used for the treatment of gonorrhea. Fifth, we assumed that the main public health response to the increase in the prevalence of resistance to an antibiotic is to switch to a new drug. In practice, however, other approaches might be employed such as increasing the dosage of the antibiotic in the first-line therapy. Finally, we assume that an individual's risk of infection depends on the local prevalence of infection but in reality, some individuals might be at an additional risk of infection depending on the extent to which they contact with individuals with other regions [22].

Our model did not differentiate anatomic sites of infection. Incorporating the emergence and transmission of AMR gonorrhea by sites of infection is challenged by the lack of data on the prevalence of AMR gonorrhea at each of the sites, fitness differences by site, and transmission rates [23]. We also assumed complete adherence to the first-line treatment guidelines (determined locally or nationally). While the actual treatment regimens used in the population may differ from the recommended guidelines, recent studies estimate adherence to the CDC guideline for the treatment of gonorrhea to be around 80% [3,24]. We assumed a homogenous risk of infection among the members of the MSM population which does not differ across individuals based on, for example, the number of sexual partners and/or condom use. Relaxing these assumptions could improve the accuracy of projections made by our model, but it is not expected to significantly affect the comparative evaluation of strategies considered here.

Implementing spatially-adaptive strategies to inform guidelines for the treatment of gonorrhea may be challenging. First, actionable surveillance of AMR gonorrhea may not be conducted in every geographic region, thereby requiring a policy for how best to inform guidelines for regions lacking surveillance. Second, revising the treatment guidelines at the local level requires establishing and maintaining a standardized mechanism for communication between local health providers and policy makers monitoring the local trend in the spread of AMR gonorrhea. Such a mechanism is essential to ensure the adherence of local health providers to the most recent treatment recommendations. Third, in 2017, GISP collected isolates from STD clinics affiliated with 27 state or city health departments. Evidence is lacking about the optimal number and geographic location of these sites if estimates of resistance prevalence provided by these sites are used to inform local treatment guidelines. Finally, enhancing surveillance systems to expand the geographic location of surveillance sites and to enable more frequent reporting and evaluation of more gonococcal isolates would increase the cost of surveillance. The value of strategies proposed here should be investigated through cost-effectiveness analyses.

In the future, the availability of point-of-care tests that also determine susceptibility to different antibiotics could inform the selection of antibiotics for the treatment of gonorrhea [25,26]. Modeling studies show that these tests could slow the spread of AMR gonorrhea and extend the usefulness of existing antibiotics for the treatment of gonorrhea [17,25,27]. Until the widespread use of these tests, however, we need to continue improving our decision-making in determining empiric treatment guidelines. While the feasibility and cost-effectiveness of these proposed changes need to be studied, the analysis presented here highlights the importance of robust local surveillance systems to slow the spread of antibiotic-resistant strains and to minimize the burden of gonorrhea. We demonstrated that using the data from surveillance programs that can distinguish local variability in AMR may prolong the effective lifespan of antibiotics without increasing the burden of the disease.

## Supporting information

**S1 Text. Additional model details and results of sensitivity analyses.** Table A: Estimated population of men who have sex with men (MSM) and the rate of reported gonorrhea cases per 100,00 MSM population in 16 U.S. metropolitan areas. Table B: Prior distributions and posterior intervals of model parameters that are assumed to be the same across all metropolitan areas. Table C: Uncertainty range and feasible intervals of model parameters related to resistance emergence and spread (assumed to be the same across all metropolitan areas). Table D: Prior distributions and posterior intervals for initial gonorrhea prevalence and initial proportion of gonorrhea that are symptomatic in each metropolitan area. Table E: Uncertainty range and feasible intervals for the initial prevalence of resistance to Drug A or Drug B in each metropolitan area. Table F: Uncertainty range and feasible intervals for the annual importation rate of cases resistant to Drug A or Drug B in each metropolitan area Table G: Prior distributions selected in the primary and sensitivity analyses Table H: Uncertainty ranges selected in the primary and sensitivity analyses Table I: Correlation between select model input parameters and the effective lifespan of drugs A and B under the 'Base' strategy. Table J: Correlation between select model input parameters and the change in the effective lifespan of drugs A and B under the 'Spatial' strategy with respect to the 'Base' strategy. Fig A: Expanded model of gonorrhea transmission among the MSM population in 16 metropolitan areas in the United States. Fig B: Behavior of function $\gamma(t)$ (defined in Eq (2)) over time. Fig C: The rate of gonorrhea cases per 100,000 MSM population in 100 simulated runs compared with the estimated rate of gonorrhea cases among the MSM population in 2017 (as shown by green dot). Fig D: The prevalence of gonorrhea among the MSM population of the metropolitan areas included in our model. Fig E: Number of gonorrhea cases treated successfully with Drugs A or B, and Drug M during the simulation year. Fig F: The estimated proportion of cases resistant to Drugs A, B or both when 200 annual gonorrhea cases are tested for drug resistance in each region during each simulation. Fig G: Posterior distribution and the correlation between select key model parameters listed in Table B and Table C. Fig H: Comparing the performance of policies in Table 1 with respect to the current policy over a 35-year simulation window. Fig I: Comparing the performance of policies in Table 1 with respect to the current policy using the recalibrated model. Fig J: Comparing the performance of policies in Table 1 with respect to the current policy using the recalibrated model where wider prior distributions and uncertainty ranges are selected (Table G-Table H). Fig K: Comparing the performance of policies in Table 1 with respect to the current policy using when the prevalence of resistance to Drug B is zero. Fig L: Displaying 200 simulated trajectories from the calibrated model where the value of transmission parameter is increased by 10%. Fig M: Comparing the performance of policies in Table 1 with respect to the current policy using when the value of the transmission parameter

is increased by 10%
(PDF)

## Acknowledgments

Data used for modeling came from CDC's Gonococcal Isolate Surveillance Project (GISP), GISP participating sites and GISP regional laboratories.

## Author Contributions

**Conceptualization:** Reza Yaesoubi, Ted Cohen, Katherine Hsu, Thomas L. Gift, Joshua A. Salomon, Yonatan H. Grad.

**Data curation:** Reza Yaesoubi, Sancta B. St. Cyr.

**Formal analysis:** Reza Yaesoubi.

**Funding acquisition:** Ted Cohen, Joshua A. Salomon.

**Investigation:** Reza Yaesoubi, Yonatan H. Grad.

**Methodology:** Reza Yaesoubi, Yonatan H. Grad.

**Software:** Reza Yaesoubi.

**Supervision:** Ted Cohen, Katherine Hsu, Thomas L. Gift, Joshua A. Salomon, Yonatan H. Grad.

**Validation:** Reza Yaesoubi.

**Visualization:** Reza Yaesoubi.

**Writing – original draft:** Reza Yaesoubi, Yonatan H. Grad.

**Writing – review & editing:** Reza Yaesoubi, Ted Cohen, Katherine Hsu, Thomas L. Gift, Sancta B. St. Cyr, Joshua A. Salomon, Yonatan H. Grad.

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
