## [Decision Letter · Decision Letter 0]

16 Aug 2021

Dear Dr. Yaesoubi,

Thank you very much for submitting your manuscript "Evaluating spatially adaptive guidelines for the treatment of gonorrhea to increase the effective lifespan of antibiotics" for consideration at PLOS Computational Biology.

As with all papers reviewed by the journal, your manuscript was reviewed by members of the editorial board and by several independent reviewers. In light of the reviews (below this email), we would like to invite the resubmission of a significantly-revised version that takes into account the reviewers' comments.

We cannot make any decision about publication until we have seen the revised manuscript and your response to the reviewers' comments. Your revised manuscript is also likely to be sent to reviewers for further evaluation.

Sincerely,

Dominik Wodarz

Associate Editor

PLOS Computational Biology

Rob De Boer

Deputy Editor

PLOS Computational Biology

Reviewer's Responses to Questions

**Comments to the Authors:**

Reviewer #1: Notes on ‘Evaluating spatially adaptive guidelines for the treatment of gonorrhea to increase the effective lifespan of antibiotics’

This paper assesses whether localised gonorrhea treatment guidelines based on regional estimates of resistance prevalence could be useful in combating the development of antibiotic resistance and thereby extending the clinical lifespan of antibiotics. Overall the paper is a timely and well written modelling analysis on a topic of great importance, namely: preserving the utility of antibiotics against Neisseria gonorrhoeae.

The authors adapt a previously-published individual-based model, which they fit to the US gonorrhea epidemic in men who have sex with men, calibrating to regional and national incidence, national prevalence and the proportion of cases presenting with symptoms in 2017. The model is clearly described and well-documented in the supplementary material. The authors use their calibrated model to perform simulation analyses to assess the impact of regional treatment guidelines on both the burden of gonorrhea and the lifespan of gonorrhea treatments (based on the development of antibiotic resistance).

The idea of regional treatment guidelines is novel and important, and the methodology of the analysis is sound. However, I have some concerns about implicit assumptions in the model calibration, namely the assumption of endemic equilibrium prevalence and incidence given the current rapidly increasing epidemic of gonorrhea. Further, some of the claims in the abstract, particularly around the lifespan extension achieved, are overstated as they rely on increasing the threshold of resistance at which drugs are abandoned (from 5% to e.g. 6.5%) which seems unlikely to be implemented in practice.

I have detailed my comments and suggestions below.

Model Calibration:

1. The authors calibrate to constant incidence over a period of 10 years (at least this is what the equations for L2 and L3 seem to suggest – I may have misunderstood) based on 2017 data assuming 5,548 cases per 100,000. Are these 10 years 2007 – 2017? Because over that time period gonorrhea incidence has increased by around 90%. Do the authors’ conclusions still hold in an increasing epidemic?

2. The authors calibrate to a prevalence of 4.5% (3.6%-5.4%) in 2017 for the whole USA (L1). The reference used to support this assumption ( Johnson Jones ML, Chapin-Bardales J, Bizune D, Papp JR, Phillips C, Kirkcaldy RD, et al. Extragenital Chlamydia and Gonorrhea Among Community Venue-Attending Men Who Have Sex with Men - Five Cities) includes a breakdown of prevalence by city, all of which are locations modelled in the manuscript, which suggests prevalence varies substantially by location: e.g. 1.8% (0.5% - 3.1%) San Francisco vs 6.7% (4.4% - 9.0%) in Houston. Is it possible to fit to these prevalence estimates in the locations presented? I note that this will likely be hard to achieve, since the prevalence and incidence estimates may be irreconcilable: in Table S1 San Francisco has much higher incidence than Houston (7,931 cases per 100,000 vs 4,858).

3. In calculating their 2017 regional incidence estimates for model calibration (Table S1), the authors assume that the proportion of all gonorrhea cases attributable to MSM is the same in each location– is there any evidence to support the validity of this assumption? The fact that the inferred incidence seems to contradict the prevalence reported by Johnson Jones et. al, as noted above, suggests there may be underlying heterogeneity.

Simulation study

4. Methods pg3: Using a t-distribution to calculate binomial proportion confidence intervals is not best practice due to the possibility of the bounds overshooting the unit interval. It would be better to use Wilson’s method which results in a 95% CI for estimating 5% prevalence is 2.7%-9.0% for a sample of 200 (vs 2.0%-8.0%) and 3.7%-6.7% for a sample of 800 (vs 3.5%-6.5%).

5. Measure of number of additional cases treated is essentially averaged over whole duration of simulation (i.e. incidence is perfectly flat)

6. The authors’ conclusion that the lifespan of a drug could be extended by 2.71 years rely on moving the resistance threshold as well as switching to a new prescribing strategy (Fig 4A), however while discussed this aspect is not emphasised, particularly in the abstract. Is it likely that the resistance threshold would be increased in practice? The policy would be very difficult to communicate and seems to tacitly assume that maintaining the current level of gonorrhea as a key goal. This is problematic in itself as the authors’ simulations assume the epidemic of gonorrhea is at equilibrium (i.e. stable) whereas gonorrhea rates have been increasing by around 5% each year in US MSM. The more realistic interpretation of the work is that the policies considered could modestly extend the lifespan of the drugs (< 1yr), but that their main impact would be reducing the annual number of cases and the burden of gonorrhea (if the current 5% threshold were kept in place), and that the lifespan could be increased further, but that this would require a reformulation of the threshold rule.

Minor points:

1. Fig1 is an exact replica of Fig1 in Yaesoubi2020, re-publishing the figure improves the readability of the paper, but I recommend acknowledging the fact and citing in the caption.

2. Pg9: Operationalizing -> Implementing

3. Pg9: Therefore, for regions without a surveillance site, it would need to be decided that data from which surveillance site(s) should be used to inform treatment guidelines

4. Pg9: optimality of the number –> the optimal number

5. S3 pg5: replies -> relies

Reviewer #2: The authors use a mathematical model of the spread of gonorrhea to explore whether locally determined rather than national empiric guidelines would increase the lifespan of antibiotics.

It was unclear to me whether such a complex model was needed to answer this question. Moreover, it was unclear whether the results (5% more could be treated, 284/5548) were really that supportive of the strategy which presumably comes with substantial local surveillance and guideline formation costs.

Abstract

- An aim is to explore the spread of drug-resistant gonorrhea, but there is no conclusion related to this

Introduction

- How long does it usually take to get DSTs? How long is the empiric prescribing time? Isn’t there an issue with such sexually transmitted pathogens of losing a patient – so how often does a switch actually happen?

Methods

- Resistance to the no longer used drugs may increase due to bystander selection no necessarily due to a decreased fitness cost.

Figure 3:

“This figure demonstrates that our simulation model can describe the geographical variation in the trends of ciprofloxacin resistant gonorrhea observed in GISP.”

I’m not sure that this figure does show this.

**Have the authors made all data and (if applicable) computational code underlying the findings in their manuscript fully available?**

Reviewer #1: None

Reviewer #2: Yes

PLOS authors have the option to publish the peer review history of their article (what does this mean?). If published, this will include your full peer review and any attached files.

Reviewer #1: No

Reviewer #2: No
---

## [Decision Letter · Decision Letter 1]

13 Dec 2021

Dear Dr. Yaesoubi,

Thank you very much for submitting your manuscript "Evaluating spatially adaptive guidelines for the treatment of gonorrhea to reduce the incidence of gonococcal infection and increase the effective lifespan of antibiotics" for consideration at PLOS Computational Biology.

As with all papers reviewed by the journal, your manuscript was reviewed by members of the editorial board and by several independent reviewers. In light of the reviews (below this email), we would like to invite the resubmission of a significantly-revised version that takes into account the reviewers' comments.

Please note that "Reviewer 3" was added to the evaluation process. The reason is that Reviewer 2, who had substantial reservations about the original version of the manuscript, was not available to comment on the revision. Reviewer 3 makes a number of comments that appear useful for the further revision of the manuscript, while generally being positive. Please take into account the points made by Reviewer 3 in your re-submission.  

We cannot make any decision about publication until we have seen the revised manuscript and your response to the reviewers' comments. Your revised manuscript is also likely to be sent to reviewers for further evaluation.

Sincerely,

Dominik Wodarz

Associate Editor

PLOS Computational Biology

Rob De Boer

Deputy Editor

PLOS Computational Biology

Reviewer's Responses to Questions

**Comments to the Authors:**

Reviewer #1: .

Reviewer #3: Review

The authors investigate how spatially informed guidelines for gonorrhea treatment can improve the lifespan of antibiotics and the number of gonorrhea cases. To this end, they propose an individual-based epidemiological and evolutionary model describing the incidence of gonorrhea in several American cities. They describe symptomatic vs. asymptomatic infections, several susceptible and resistant gonorrhea genotypes, and treatment by three possible antibiotics (A, B, last-line). They derive plausible sets of parameters reproducing well the overall prevalence & yearly incidence of gonorrhea in the USA, fraction symptomatic, and for which drug resistance for both A and B reaches at least 5% over the 50 years of the simulation.

The manuscript is interesting and although the benefits of a spatially informed strategy appear weak, it is still a worthwhile exploration.

My major comments are the following:

1) Clarity of the model description can be improved. Specific suggestions:

- Clarify the structure of treatment, particularly under what circumstances individuals move to drug A, drug B, drug M, and what happens in case of treatement success vs. failure. Figure 1 is not very informative in this respect. I don't find the diagram of fig. S1 very clear in terms of the type of treatment applied depending on stage, resistance status, symptoms. (e.g. Asymptomatic can seek second line treatment?). Please improve this. Paragraph S1.2 need to specify in more details what happens to individuals on adequate / inadequate treatment.

- Notation must be changed to improve consistency and clarity. e.g. S is used for number of susceptibles and for S_0 (total number of cases successfully treated); L_0 is used for lifespan but L_1, L_2, for likelihood components (in S3.1). N is used for number treated successfully and for total population size (in S1.2). It is also used for number of simulations in S3.5. Paragraph S3.3, notation problem again, beta used for proportion of cases that are symptomatic (and for transmission rate).

- "Results", please indicate what parameters are varied across simulations and inferred (at least the most important ones)

2) The benefits of spatial strategy explored here of course depends on the spatial heterogeneity in frequency of resistance. This is acknowledged in Discussion. The spatial heterogeneity in resistance is mentioned in "Results". I think this could be emphasised and explained further. The authors could add 1-2 sentences to explain what processes in their model generates heterogeneity in the frequency of resistance (in the Results). For example, variability in the initial frequency of resistance could be a driver of spatial heterogeneity in frequency, is this effect important here? The authors could also plot (as an extra-panel of figure 3 for example) a measure of spatial variation in the frequency of resistance (e.g. variance in frequency as a function of time) in the data, and in the simulations. This would allow a more precise comparison of the extent of spatial heterogeneity in the data and in the simulations.

Minor comments:

Page 5 need to specify "will seek retreatment with some delay" -> retreatment with the alternative drug, correct? Please specify

Page 5 need to give a bit more details on the shape of the declining cost of resistance, and the timescale at which it declines

Page 6 might be helpful to plot N_A(t), N_B(t) and N_M(t) to give a better intuition of the effective lifespan measure.

Page 6 I was not sure why compute the expected number of additional cases of gonorrhea successfully treated as S_0 * ∆L / L_0. Why not compute it directly from the difference in the results of simulations of the "Base" vs. other strategies? (or, is it actually the same? please specify)

Page 7 please define the intervals given in parentheses e.g. "200 (169, 232)"

Page 10 "Incorporating the exact prevalence of gonorrhea in each city could improve the predictive power of the model but is not expected to influence the comparative performance of strategies simulated here" -> I believe variability in prevalence across cities (particularly if correlated with frequency of resistance) could change the quantitative impact of the different strategies (figure 4). This seems important to note, as the quantitative effects are an important part of the results.

Not clear about the role of diagnosis of symptomatic / asymptomatic cases in relation to the model as explained in S1.2. Is "diagnosed" equivalent to "waiting to receive first line treatment"?

S1.2 typo in the list of possible k ("m16" instead of ",16")

Why isn't the estimated fraction of infections that are symptomatic (49.2%, table S2) close to the fraction of symptomatic in the data (67.9%)?

Typo at the end of S3.4 "because that" (remove "that")

S3.5 I don't get the point of removing L_max in the expression from the weight, doesn't exp(-L_{max}) cancel out? If the authors agree, then there just remains weight proportional to L_i / sum(L_i).

I do not understand the difference between prior / posterior distributions of parameters, versus "uncertainty range" / "feasible interval" used for parameters related to emergence of resistance (table S3). The criterion of reaching at least 5% resistance can also be formulated as a pseudo-likelihood function with a particular form (i.e. probability 1 if 5% resistance is reached for both drugs, 0 otherwise).

**Have the authors made all data and (if applicable) computational code underlying the findings in their manuscript fully available?**

Reviewer #1: Yes

Reviewer #3: **No: **Authors declared code will be put on Github

PLOS authors have the option to publish the peer review history of their article (what does this mean?). If published, this will include your full peer review and any attached files.

Reviewer #1: No

Reviewer #3: No
---

## [Editor Report · Decision Letter 2]

16 Jan 2022

Dear Dr. Yaesoubi,

We are pleased to inform you that your manuscript 'Evaluating spatially adaptive guidelines for the treatment of gonorrhea to reduce the incidence of gonococcal infection and increase the effective lifespan of antibiotics' has been provisionally accepted for publication in PLOS Computational Biology.

Best regards,

Dominik Wodarz

Associate Editor

PLOS Computational Biology

Rob De Boer

Deputy Editor

PLOS Computational Biology

---

## [Editor Report · Acceptance letter]

4 Feb 2022

PCOMPBIOL-D-21-01173R2 

Evaluating spatially adaptive guidelines for the treatment of gonorrhea to reduce the incidence of gonococcal infection and increase the effective lifespan of antibiotics

Dear Dr Yaesoubi,

I am pleased to inform you that your manuscript has been formally accepted for publication in PLOS Computational Biology. Your manuscript is now with our production department and you will be notified of the publication date in due course.

With kind regards,

Andrea Szabo
